# PRINCIPAL COMPONENT ANALYSIS FOR CROSS-SECTIONALLY CORRELATED PRICING ERRORS

## ABSTRACT

We propose a new estimator for factor pricing models which we refer to as Principal Component Analysis for Cross-Sectionally Correlated Pricing Errors (PCA-XC). Our estimator aims to find the factor pricing model that well explains the time-series variation of asset returns and well handles the correlations of cross-section of pricing errors that we present exist in real-world data. The proposed estimator is defined by a new regularized minimization problem in which finding a solution is difficult. This contrasts with other related estimators whose corresponding minimization problem admits an analytic solution. To this end, we propose an approximate algorithm that solves our proposed minimization problem based on the alternating least squares method.

## 1 INTRODUCTION

In this paper, we study one of the central problems in the field of finance, namely, the estimation of the multifactor model for asset pricing that explains how prices of various assets, *e.g.*, stocks and bonds, are set to the current values by using a small number of factors compared to the number of assets. In this section, we explain the importance of the estimation problem and discuss how the problem can be cast into a problem of unsupervised learning. To begin with, we describe factor pricing models in its simplest form as follows.

**Factor pricing models** Let us consider a bivariate linear mapping

$$E(R^{ei}) = \beta_i E(f) + \alpha_i \ \text{ for } \ i \in \{1, 2, \cdots, N\} \tag{1}$$

from a pair of real numbers $(\alpha_i, \beta_i)$ to a real number $E(R^{ei})$ where a natural number $N$ denotes the number of all assets and $E(\cdot)$ denotes the expectation operator defined on the set of all possible random variables $R^{ei}$ and $f$ that represent excess return of asset $i$ and *pricing factor*, respectively. The second term in Eq. (1), $\alpha_i$, represents the *pricing error*. Moreover, let us consider $N$ constraints where $\alpha_i$ in Eq. (1) is equal to zero for all $i$, *i.e.*,

$$\alpha_i = 0 \ \text{ for } \ i \in \{1, 2, \cdots, N\}. \tag{2}$$

Here, we explicitly write the constraint on $\alpha_i$'s instead of directly applying it to Eq. (1) in order to clarify that finding a model with smaller pricing errors, $\alpha_i$'s, is one of the two objectives that an estimator proposed in this paper pursues. Given a set of realizations $R_1^{ei}, R_2^{ei}, \cdots, R_T^{ei} \in \mathbb{R}$ of the random variable $R^{ei}$ generated at time $t \in \{1, 2, \cdots, T\}$ for every $i \in \{1, 2, \cdots, N\}$, the estimation of our interest consists of finding $\beta_i$ and $E(f)$ that satisfy Eqs. (1, 2) as closely as possible.

Figure 1 graphically illustrates Eq. (1) without imposing the constraints of Eq. (2). Each of the dots in the figure corresponds to an arbitrary asset $i$ that is located in the $(E(R^e), \beta)$ coordinate system based on the expected excess return $E(R^{ei})$, the *factor loading* $\beta_i$ and the pricing error $\alpha_i$. The expected value $E(f)$ of the pricing factor, called the *factor risk premium*, serves as the slope of the straight line which, for example, corresponds to the Security Market Line in the Capital Asset Pricing Model developed by Sharpe (1964) and Lintner (1965).

The factor pricing model, described by Eqs. (1, 2) or the straight line in Figure 1, postulates that an asset $i$'s expected excess return linearly depends on the factor loading $\beta_i$. The straight line exhibits how the factor pricing model values the expected excess return of an arbitrary asset given the asset's

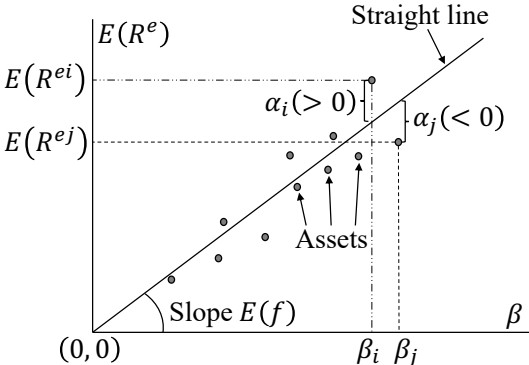

Figure 1: Illustration of the expected return-beta representation. Each of the dots corresponds to an arbitrary asset $i$. This figure is modified from Figure 12.1 of Cochrane (2005).

factor loading. If we acknowledge that the deviation of the dots from the straight line is significant, we could decide either of two things. First, the factor pricing model is wrong. For example, the underlying factor $f$ is chosen in a wrong way. This leads us to conclude that a better pricing factor should be chosen or more pricing factors be added to the model. That is, the model needs to be improved. Second, the world is wrong. This means that the dots do not line up on the straight line since the corresponding assets are "mis-priced." In Figure 1, for example, the factor pricing model undervalues (resp. overvalues) the expected excess return of asset $i$ (resp. $j$), *i.e.*, $\alpha_i > 0$ (resp. $\alpha_j < 0$). This might present practical trading opportunities for shrewd investors (Cochrane, 2005). This is one of many reasons that correctly estimating the underlying pricing factor is a central problem in the field of finance that led to more than 300 published pricing factor candidates (Harvey et al., 2016).

**Definition of multifactor models** In this paper, we consider multifactor models for the following reason. Note first that we can generalize the factor pricing models described above by adding more pricing factors. It is therefore more sensible to use $K(> 1)$ pricing factors instead of a single factor to explain expected excess returns of a large number of arbitrary assets (Fama & French, 1993; 2015). The multifactor models amount to a $(K + 1)$-variate linear mapping

$$E(R^{ei}) = \beta_{i,1}E(f^1) + \beta_{i,2}E(f^2) + \cdots + \beta_{i,K}E(f^K) + \alpha_i \text{ for } i \in \{1, 2, \cdots, N\} \quad (3)$$

from a tuple of real numbers $(\alpha_i, \beta_{i,1}, \cdots, \beta_{i,K})$ to a real number $E(R^{ei})$ where a natural number $K$ denotes the number of the pricing factors that are used to model the expected excess returns across assets, or the cross-section of expected excess returns. Usually, $K$ is set to be much smaller than $N$.

Let $f = [f^1, f^2, \cdots, f^K]^\top$ and $\beta_i = [\beta_{i,1}, \beta_{i,2}, \cdots, \beta_{i,K}]^\top$ be $K \times 1$ column vectors of the pricing factors and the factor loadings, respectively. (We let $A^\top$ denote the transpose of a matrix $A$.) If Eqs. (2) and (3) are satisfied and the $\beta_i$ is computed as

$$\beta_i = \Sigma_f^{-1} cov(f, R^{ei}) \text{ for } i \in \{1, 2, \cdots, N\} \quad (4)$$

where $\Sigma_f$ is the $K \times K$ covariance matrix of $f$ and $cov(f, R^{ei})$ denotes the $K \times 1$ column vector whose $k$-th entry is $cov(f^k, R^{ei})$, then we say that there is a multifactor model with factors $f^1, f^2, \cdots, f^K$ (Back, 2017, Section 6.2). We refer to the multifactor model described by Eqs. (2–4) as the $K$ factor model.

**Unsupervised learning problem for the multifactor models** The main objective of this study is to propose a new method for estimating the pricing factors and the factor loadings of the multifactor model based on a given set of training data $D = \{R_t^{ei} : t = 1, 2, \cdots, T, i = 1, 2, \cdots, N\}$. The estimator aims to provide the $T$ estimates of the pricing factors $\hat{f}_1, \hat{f}_2, \cdots, \hat{f}_T \in \mathbb{R}^{K \times 1}$ and the $N$ estimates of the factor loadings $\hat{\beta}_1, \hat{\beta}_2, \cdots, \hat{\beta}_N \in \mathbb{R}^{K \times 1}$ that fit well to the conditions in Eqs. (2–4). Note that the estimator studied in this paper has nothing to do with predicting, say, an expected excess return, from unseen data. In other words, the estimation problem studied in this paper is an unsupervised learning problem.

**Notation**  We summarize notation used in this paper as follows. Let $N, T$ and $K$ be the number of all assets, the number of time-series observations, and the number of factors, respectively. We assume that $K < N$ and $K < T$ throughout the paper. Let $m, n \in \mathbb{N}$. $I_n$ is an $n \times n$ identity matrix. $\mathbb{1}_n$ is a $n \times 1$ column vector of ones. For a matrix $A \in \mathbb{R}^{m \times n}$, we let $rank(A)$ denote the rank of $A$, $A^\top$ its transpose, $\|A\|_F$ its Frobenius norm, and $vec(A) \in \mathbb{R}^{mn \times 1}$ the vec operation applied to $A$. For a square matrix $S$, $tr(S)$ denotes the trace of $S$. For matrices $A$ and $B$, $A \otimes B$ denotes the Kronecker product of $A$ and $B$. We let $\mathbb{S}_+^n$ denote the set of all $n \times n$ symmetric positive-semidefinite matrices. For a $V \in \mathbb{S}_+^n$, we define $\| \cdot \|_V : \mathbb{R}^n \to [0, \infty)$ by $\|x\|_V = \sqrt{x^\top V x}$ for all $x \in \mathbb{R}^{n \times 1}$. The set of training data $D$ is compactly represented by a $T \times N$ matrix $X$ whose $(t, i)$-th entry is $R_t^{ei}$. We define a $T \times T$ matrix $P_1 = \frac{1}{T} \mathbb{1}_T \mathbb{1}_T^\top$ which is a projection matrix onto the linear subspace spanned by $\mathbb{1}_T$. We define a $T \times T$ matrix $M_1 = I_T - P_1$ that annihilates the component that is parallel to the subspace spanned by $\mathbb{1}_T$, *i.e.*, $\mathbb{1}_T^\top (M_1 x) = 0$ for all $x \in \mathbb{R}^{T \times 1}$. We widely use the fact that $P_1$ and $M_1$ are symmetric and idempotent.

## 2 MOTIVATION

In this section, we elucidate a critical issue that motivates the estimator proposed in this paper, supported by preliminary experiments conducted using real-world data (French, 2022). We consider the three factor model described by the time-series regression

$$R_t^{ei} = \alpha_i + \beta_{i,\texttt{Mkt-RF}} R_t^{\texttt{Mkt-RF}} + \beta_{i,1} f_t^1 + \beta_{i,2} f_t^2 + \epsilon_t^i \text{ for } t \in \{1, 2, \cdots, T\} \qquad (5)$$

of excess returns $R_t^{ei}$ on the pricing factors $(R_t^{\texttt{Mkt-RF}}, f_t^1, f_t^2)$. We fix one of the three factors as the market's excess return $R_t^{\texttt{Mkt-RF}}$ that represents the entire US stock market and choose the remaining two from five candidates, $\{\texttt{SMB}, \texttt{HML}, \texttt{CMA}, \texttt{RMW}, \texttt{Mom}\}$, which account for $\binom{5}{2} = 10$ combinations. We consider $N = 25$ portfolios formed on size and book-to-market equity ratio, often called the $5 \times 5$ Size-B/M portfolios, as test assets indexed by $i \in \{1, 2, \cdots, 25\}$. It is known that they are well explained when $(f^1, f^2) = (\texttt{SMB}, \texttt{HML})$ (Fama & French, 1993). For each $i$, we run the time-series regression from January 2017 to December 2021 (60 months) and obtain estimates of the factor loadings $(\hat{\beta}_{i,\texttt{Mkt-RF}}, \hat{\beta}_{i,1}, \hat{\beta}_{i,2})$, the pricing error $\hat{\alpha}_i$ and the residuals $\hat{\epsilon}_t^i$.

A caveat is that the estimation problem described in the previous section does not exactly match this time-series regression analysis. In the former case, both factors and regression coefficients are estimated, whereas in the latter, it relies on predefined factors without considering their estimation. By fixing one element for estimation and utilizing the established knowledge of explanatory power exhibited by various combinations of factors, we gain a more intuitive understanding of the essential characteristics that "appropriate" factors and, consequently, factor models necessarily possess. These characteristics form the basis for designing the criterion of the proposed factor model estimator.

Figure 2 (left) illustrates the absolute value of the sample correlation coefficients between $\hat{\epsilon}_1^i, \cdots, \hat{\epsilon}_T^i$ and $\hat{\epsilon}_1^j, \cdots, \hat{\epsilon}_T^j$ that are identified by the brightness of the $(i, j)$-th grid with the color bar. Figure 2 (right) presents the average absolute values of the sample correlation coefficients and the Bartlett (1951)'s sphericity test statistic for the hypothesis that the correlation matrix equals the identity matrix. The figure reveals that the choice of $(f^1, f^2) = (\texttt{SMB}, \texttt{HML})$ indeed results in the smallest correlation. This is shown in the top-left subfigure of Figure 2-(a), represented by the darkest surface. It is also observed by the smallest values presented by the leftmost two bars in Figure 2-(b). The figure demonstrates that excluding either $\texttt{SMB}$ or $\texttt{HML}$ leads to a rise in cross-sectional correlation, while excluding both $\texttt{SMB}$ and $\texttt{HML}$ further increases the cross-sectional correlations. These observations suggest that "appropriately" chosen pricing factors well regress out covariations in the test assets' returns, making the unexplained part of the test assets, *i.e.*, $\hat{\alpha}_i + \hat{\epsilon}_t^i$, less likely to be correlated across assets. On the other hand, in the opposite situation where the chosen pricing factors are "inappropriate" to explain the test assets, it is more likely that $\hat{\alpha}_i + \hat{\epsilon}_t^i$ *is cross-sectionally correlated*.

Meanwhile, in the context of evaluating an estimated factor model, it is common to test the null hypothesis $H_0 : \alpha_1 = \cdots = \alpha_N = 0$ by utilizing a test statistic of the form:

$$q = c \, \hat{\alpha}^\top \hat{W}^{-1} \hat{\alpha} \qquad (6)$$

where $\hat{\alpha} = [\hat{\alpha}_1, \cdots, \hat{\alpha}_N]^\top$, $\hat{W}$ is the estimated residual covariance matrix and $c$ is a constant that does not depend on $\hat{\alpha}$ and $\hat{W}$ (Fama & MacBeth, 1973; Gibbons et al., 1989). Evaluation of the

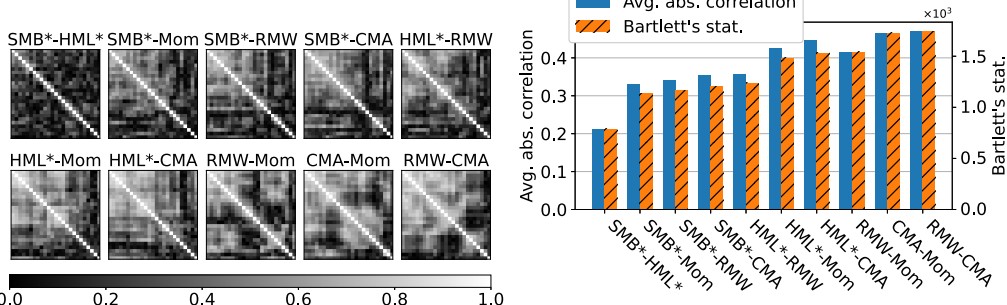

Figure 2: Statistics for residuals of $5 \times 5$ Size-B/M portfolio returns regressed on three factor models. One factor is fixed to the market's excess return and the remaining two are written in subfigure title (left) and on the $x$-axis (right). Factors with $*$ are related to the test assets.

model is conducted by checking whether $|q| \leq \delta$ for a predefined threshold $\delta$ in which case it is concluded that the factor model is correctly estimated. Conversely, if $|q| > \delta$, it is inferred that the factor model is not correctly estimated. Therefore, it is important to find a model capable of achieving a small value of $|q|$ that passes the test for the null $H_0$. As shown in Figure 2, however, we note that $\hat{W}$ can significantly deviate from $I_N$, particularly when "inappropriate" factors are selected. In such cases, the computed value of $q$ under the assumption of $\hat{W} = I_N$ might exhibit highly unpredictable difference from the value obtained without this assumption. This observation motivates the incorporation of $q$ and $\hat{W}$ into the estimation criterion such that models with smaller $|q|$ are preferred.

## 3 RELATED WORK AND OUR CONTRIBUTIONS

Arbitrage Pricing Theory of Ross (1976) pioneered a line of research where statistical factor structure in the covariances of excess returns between assets are considered first, from which it is derived that the cross-section of expected excess returns is explained by the multifactor model (Eqs. (2–4)). The estimation problem of our interest where both of the pricing factors and the factor loadings are latent and should be estimated resorts to the Principal Component Analysis (PCA), justified by Chamberlain & Rothschild (1983) and Connor & Korajczyk (1986), which has been a dominant form of estimator in the literature evidenced by large number of publications in recent years (Fan et al., 2016; Kozak et al., 2018; Kelly et al., 2019; Pukthuanthong et al., 2019; Lettau & Pelger, 2020a; Giglio & Xiu, 2021; Bryzgalova et al., 2023).

The conventional PCA due to Chamberlain & Rothschild (1983) and Connor & Korajczyk (1986) removes sample mean from the data by using $\tilde{X} = M_1 X$ instead of $X$ and apply the eigen-decomposition to $\hat{\Sigma} = \frac{1}{T} X^\top M_1 X = \frac{1}{T} \tilde{X}^\top \tilde{X}$ so as to explain as much time-series variation in the de-meaned data $\tilde{X}$ as possible. This is explained in Box A in Figure 3 where the objective is to find the factors $\tilde{F} \in \mathbb{R}^{T \times K}$ and the loadings $\Lambda \in \mathbb{R}^{N \times K}$ that well approximate $\tilde{X}$. The conventional PCA does not explicitly address pricing error information represented in the Boxes B and C.

The most related work to our study is that of Lettau & Pelger (2020a,b) where risk-premium PCA (RP-PCA) estimator is proposed. They generalized the conventional PCA estimators by removing the assumption that the mean of the data matrix $X$ is equal to zero based on the observation that this assumption might be restrictive if the means have information about the factor structure. The RP-PCA differs from regularized PCA estimators proposed for other applications, *e.g.*, low-rank matrix approximation (Srebro & Jaakkola, 2003; Recht et al., 2012; Udell et al., 2016) and matrix completion (Keshavan et al., 2009; Wright et al., 2009; Candès & Tao, 2010), in that the RP-PCA adds an *economically motivated* regularization term that accounts for cross-sectional pricing errors.

The RP-PCA explicitly takes into account Box B in Figure 3 where the risk premia of the assets and the factor risk premia are compared. In other words, Box B aims to find $\bar{F} \in \mathbb{R}^{K \times 1}$ and $\Lambda \in \mathbb{R}^{N \times K}$ in such a way that it minimizes the pricing errors, which are estimated by the difference

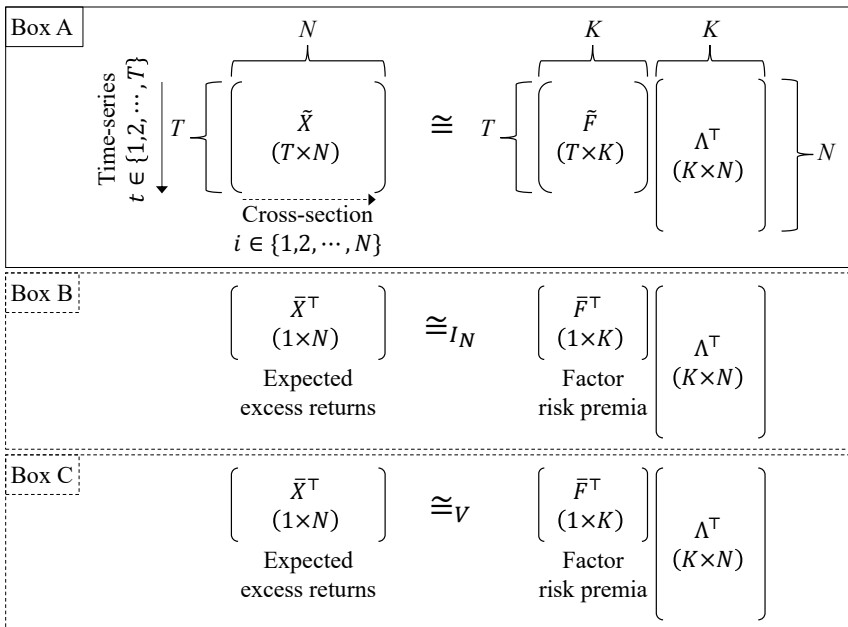

Figure 3: Estimators of the $K$ factor models. The conventional PCA aims to find $\tilde{F}$ and $\Lambda$ that well approximate the de-meaned time-series variations in $\tilde{X}$ (Box A). The RP-PCA adds a regularization term to explicitly address the pricing error (Box B). The PCA-XC extends the RP-PCA by allowing the pricing errors to be calculated in a more general way, represented by $\cong_V$ (Box C) that indicates the pricing error is measured by $\|\bar{X} - \bar{F}\Lambda^\top\|_V$ for any arbitrary $V \in \mathbb{S}^n_+$.

between $\bar{X}$ and $\bar{F}\Lambda^\top$, measured by $\|\bar{X} - \bar{F}\Lambda^\top\|_{I_N}$. By simultaneously considering the time-series variations (Box A) and the cross-section of pricing errors (Box B) in the framework of a regularized minimization problem, Lettau & Pelger (2020a;b) showed that the RP-PCA can find pricing factors that cannot be detected by the conventional PCA. They also proved that RP-PCA can estimate factors more efficiently than the conventional PCA and that it outperforms PCA in the presence of "weak" factors.

A crucial issue with their approach is the lack of explicit consideration for the real-world scenario, characterized by *correlated* pricing errors observed in Section 2. This circumstance could result in substantial differences between $\|\bar{X} - \bar{F}\Lambda^\top\|_{I_N}$ and $\|\bar{X} - \bar{F}\Lambda^\top\|_V$, with the later being the crucial distance function for pricing model estimation when $V$ equals to the precision matrix of pricing errors, as it represents the test statistic $q$ in Eq. (6). However, the method for incorporating $\|\cdot\|_V$ for a general $V$ into pricing model estimation has not been explored in the literature, despite its significance in handling real financial data.

**Our contributions**  Our contribution to the machine learning literature is twofold. First, we propose a new estimator for pricing factors and the associated factor loadings which are defined as solutions to a regularized minimization problem (Section 4.1). To this end, we restate the estimation problem for asset pricing model in a more familiar way for the machine learning community (Section 1 and Figure 3). Then, the estimates of pricing factors and factor loadings will aim to closely approximate the time-series fluctuations in the training set of excess returns, while ensuring that the cross-section of pricing errors remains jointly small, even in the presence of correlated pricing errors across assets. Specifically, we extend the RP-PCA by allowing the distance to be defined by a seminorm $\|\cdot\|_V$ for any arbitrary $V \in \mathbb{S}^n_+$, which entails the consideration of Box C in Figure 3.

Second, we provide an optimization method that approximately solves the proposed minimization problem (Section 4.2). We employ the alternating least squares method, which solves least squares problems for factor loadings (resp. factors) with factors (resp. factor loadings) fixed, iteratively (Algorithm 1). We explain that it is non-trivial to apply the alternating least squares method to the optimization problem of our interest and prove that the proposed algorithm converges and generates

well-defined iterates (Section 4.2). By doing so, we introduce a novel computational framework for examining factor pricing model estimators defined by various choices for $V$ that allows a broader and more general measurement of pricing errors.

# 4 THE PROPOSED ESTIMATOR OF THE $K$ FACTOR MODELS

In this section, we present our proposed estimator of the $K$ factor models, which is defined by an optimization problem that we propose in this paper. We explain the objective function of the proposed optimization problem and relate it to the conventional PCA estimator and the RP-PCA estimator.

## 4.1 THE PROPOSED ESTIMATOR AND MINIMIZATION PROBLEM

Based on the preliminary empirical findings in Section 2, it appears that correlations between pricing errors of distinct assets may not be negligible if inappropriate pricing factors are employed to explain asset excess returns. Drawing on these preliminary results and the concept outlined in Figure 3, we propose a novel estimator of the $K$ factor model, which we refer to as PCA-XC. The estimates of $F$ and $\Lambda$ computed through PCA-XC are determined as solutions to the following unconstrained nonconvex minimization problem.

$$(\hat{\Lambda}, \hat{F}) \in \arg\min \ \phi(\Lambda, F) := \frac{1}{NT} \left\| \tilde{X} - \tilde{F}\Lambda^\top \right\|_F^2 + \frac{\eta}{N} \left\| \bar{X} - \Lambda\bar{F} \right\|_V^2 \qquad (\mathcal{P})$$

where the objective function $\phi : \mathbb{R}^{N \times K} \times \mathbb{R}^{T \times K} \to [0, \infty)$ is specified by the training data $X \in \mathbb{R}^{T \times N}$, a matrix $V \in \mathbb{S}_+^N$ that defines the seminorm $\| \cdot \|_V : \mathbb{R}^N \to [0, \infty)$ and a regularization parameter $\eta \in [0, \infty)$. The following reformulation of $\phi$

$$\begin{aligned}
(NT)\phi(\Lambda, F) \\
= tr\left( M_1(X - F\Lambda^\top)(X - F\Lambda^\top)^\top M_1 + \eta P_1(X - F\Lambda^\top)V(X - F\Lambda^\top)^\top P_1 \right)
\end{aligned} \qquad (7)$$

is useful, which is derived in in Section B of Appendix. Note that $\phi$ in Eq. (7) consists of two terms whose trade-off is determined by $\eta$. The first term takes into account the variations of the de-meaned excess returns, $\tilde{X}$, that are not explained by the de-meaned factors $\tilde{F}$ and the loading $\Lambda$. (See Box A in Figure 3.) In other words, the first term annihilates the first-order information in the training data and measures only the second-order information, namely, the sum of squares of de-meaned residuals of the training data $X$ that are not explained by the $F$ and $\Lambda$.

On the other hand, the second term quantifies solely the first-order information, specifically, the sample mean of residuals of the training data $X$ that are not explained by $F$ and $\Lambda$. Furthermore, we allow the use of an arbitrary $V \in \mathbb{S}_+^N$ to adapt the situation where the pricing errors, proxied by $\bar{X} - \Lambda\bar{F} \in \mathbb{R}^{N \times 1}$, are cross-sectionally correlated.

If $\phi$ in Eq. (7) only has the first term, *i.e.*, $\eta = 0$, then PCA-XC reduces to the conventional PCA estimator where the pricing factors and factor loadings are estimated by computing the eigen-decomposition of $X^\top M_1 X$. If we replace $\eta$ with $1 + \gamma$ and $V$ with $I_N$, then our proposed objective function $\phi$ becomes a function $\phi_{RP} : \mathbb{R}^{N \times K} \times \mathbb{R}^{T \times K} \to [0, \infty)$ defined by

$$\begin{aligned}
(NT)\phi_{RP}(\Lambda, F) \\
= tr\left( M_1(X - F\Lambda^\top)(X - F\Lambda^\top)^\top M_1 + (1 + \gamma)P_1(X - F\Lambda^\top)(X - F\Lambda^\top)^\top P_1 \right)
\end{aligned} \qquad (8)$$

for $\gamma \in [-1, \infty)$ which is exactly the same as the objective function that defines the RP-PCA estimator of Lettau & Pelger (2020a;b). Consequently, our PCA-XC subsumes the traditional PCA and RP-PCA as special cases.

## 4.2 OPTIMIZATION ALGORITHM BASED ON THE ALTERNATING LEAST SQUARES

A critical hurdle in employing PCA-XC arises from computation. Specifically, it is difficult to solve Problem ($\mathcal{P}$) due to the existence of the matrix $V$ that accepts an arbitrary matrix in $\mathbb{S}_+^N$. In comparison, the conventional PCA (or our estimator with $\eta = 0$) and RP-PCA estimators can find an exact solution simply by applying the eigen-decomposition to the matrix in the form of

$X^\top(I_T+\frac{\gamma}{T}\mathbb{1}\mathbb{1}^\top)X$ because the first-order optimality conditions of the minimization problems of the conventional PCA and RP-PCA estimator imply that $F$ in Eq. (8) can be removed using the relation $F = X\Lambda^\top(\Lambda^\top\Lambda)^{-1}$. This substitution is impossible for our proposed minimization problem due to the existence of $V$ in the second term in Eq. (7). Section B in Appendix offers a detailed explanation for this issue.

In order to find an approximate solution to Problem ($\mathcal{P}$), we employ the alternating least squares method where $\phi(\Lambda, F)$ is minimized for one variable at a time with the other variable fixed. In the update step for pricing factors, we find the minimum of $\phi(\Lambda_*, F)$ over $F$ with a fixed $\Lambda_*$ by solving

$$
\begin{aligned}
(\Lambda_*^\top \otimes I_T)\,[I_N \otimes M_1 + \eta(V \otimes P_1)]\,\mathrm{vec}(X) \\
= (\Lambda_*^\top \otimes I_T)\,[I_N \otimes M_1 + \eta(V \otimes P_1)]\,(\Lambda_* \otimes I_T)\mathrm{vec}(F)
\end{aligned}
\tag{9}
$$

for $F$, and in the update step for factor loadings, we find the minimum of $\phi(\Lambda, F_*)$ over $\Lambda$ with a fixed $F_*$ by solving

$$
\begin{aligned}
(I_N \otimes F_*^\top)\,[I_N \otimes M_1 + \eta(V \otimes P_1)]\,\mathrm{vec}(X) \\
= (I_N \otimes F_*^\top)\,[I_N \otimes M_1 + \eta(V \otimes P_1)]\,(I_N \otimes F_*)\mathrm{vec}(\Lambda^\top)
\end{aligned}
\tag{10}
$$

for $\Lambda$. Sufficient conditions for existence and uniqueness of solutions to Eqs. (9) and (10) are given in the following proposition which is proved in Section C in Appendix.

**Proposition 4.1.** *Suppose that $V \in \mathbb{S}_+^N$. Then, there exist solutions to equations (9) and (10). If it is additionally assumed that $V$ is positive-definite, $\eta > 0$ and $\Lambda_*$ and $F_*$ have full column rank, i.e., $rank(\Lambda_*) = rank(F_*) = K$, then the solutions are unique.*

Algorithm 1 summarizes the PCA-XC that defines the estimates as $\hat{\Lambda} = \Lambda_{n_{max}}$ and $\hat{F} = F_{n_{max}}$. For iterates $(\Lambda_n, F_n)$ generated by the algorithm for any $V \in \mathbb{S}_+^N$ and $\eta \in [0, \infty)$, it is true that the sequence $\{\phi(\Lambda_n, F_n)\}$ is monotonically decreasing and so is convergent. Indeed, it is clear that the function $F \mapsto \phi(\Lambda_*, F)$ is convex for any $\Lambda_*$. Thus, $F_n$ that satisfies the first-order optimality condition (Eq. (11)) is the global minimum of the convex function $F \mapsto \phi(\Lambda_{n-1}, F)$. It similarly holds for the function $\Lambda \mapsto \phi(\Lambda, F_*)$.

---

**Algorithm 1:** Principal Component Analysis for Cross-Sectionally Correlated Pricing Errors

**input :** $X \in \mathbb{R}^{T \times N}, V \in \mathbb{S}_+^N, \eta \in [0, \infty), n_{max} \in \mathbb{N}$
**output:** $\Lambda_{n_{max}} \in \mathbb{R}^{N \times K}$ and $F_{n_{max}} \in \mathbb{R}^{T \times K}$
```
/* Initialization */
```
Choose $\Lambda_0 \in \mathbb{R}^{N \times K}$ and $F_0 \in \mathbb{R}^{T \times K}$ that satisfy $rank(\Lambda_0) = rank(F_0) = K$
**for** $n = 1, 2, \cdots, n_{max}$ **do**

> ```
> /* Update step for pricing factors */
> ```
> Define $F_n$ as a solution to Eq. (11).
>
> $$
> \begin{aligned}
> (\Lambda_{n-1}^\top \otimes I_T)\,[I_N \otimes M_1 + \eta(V \otimes P_1)]\,\mathrm{vec}(X) \\
> = (\Lambda_{n-1}^\top \otimes I_T)\,[I_N \otimes M_1 + \eta(V \otimes P_1)]\,(\Lambda_{n-1} \otimes I_T)\mathrm{vec}(F)
> \end{aligned}
> \tag{11}
> $$
>
> ```
> /* Update step for factor loadings */
> ```
> Define $\Lambda_n$ as a solution to Eq. (12).
>
> $$
> \begin{aligned}
> (I_N \otimes F_n^\top)\,[I_N \otimes M_1 + \eta(V \otimes P_1)]\,\mathrm{vec}(X) \\
> = (I_N \otimes F_n^\top)\,[I_N \otimes M_1 + \eta(V \otimes P_1)]\,(I_N \otimes F_n)\mathrm{vec}(\Lambda^\top)
> \end{aligned}
> \tag{12}
> $$

**end**

---

## 5 EXPERIMENTS

In this section, we demonstrate numerical and economic properties of PCA-XC by conducting experiments on the real-world data which span $T = 60$ months from January 2017 to December 2021. We use excess returns of $5 \times 5$ Size-B/M portfolios, *i.e.*, $N = 25$, that are represented by a $T \times N$

matrix $X$ as in Section 2. We select the regularization parameters $\eta$ of PCA-XC in Eq. (7) and $\gamma$ of RP-PCA in Eq. (8) to satisfy $\eta = 1 + \gamma$, and solely display the value of $\eta$ to ensure a fair comparison between the two estimators since, by doing so, they impose the same amount of importance to the regularization terms. We fix $K = 4$ as in the simulation study of Lettau & Pelger (2020a).

**Numerical properties** We demonstrate the numerical properties of Algorithm 1 with a fixed data matrix $X \in \mathbb{R}^{T \times N}$ and a regularization parameter $\eta = 10$, while varying the input $V \in \mathbb{S}_+^n$ to handle cross-sectional correlation in the pricing errors. First, we consider the case when $V = I_N$ where the PCA-XC is reduced to the RP-PCA, and so the exact solution can be found. Let us denote the exact solution by $(\Lambda_*, F_*)$ and the minimum value of the objective function by $\phi_* = \phi(\Lambda_*, F_*)$. For a set of iterates $\{(\Lambda_n, F_n) : n = 0, 1, \cdots, n_{max}\}$ created by Algorithm 1, let us define $\phi_n = \phi(\Lambda_n, F_n)$. We run the algorithm five times with five different random initializations $(\Lambda_0, F_0)$, and exhibit the suboptimality measured by $\phi_n - \phi_*$ and the distance between the iterates and the exact solution $\|F_n \Lambda_n^\top - F_* \Lambda_*^\top\|_F$ in the subfigures (a) and (b) of Figure 4 where we can clearly see that the suboptimality and the distance both converge to zero[1] for every random initialization.

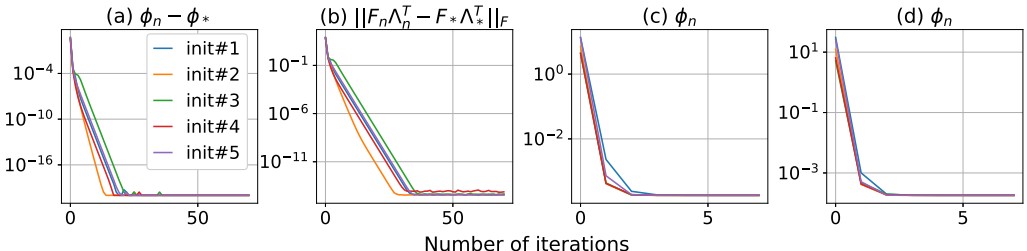

Figure 4: Convergence of Algorithm 1. (a) and (b) display results for $V = I_N$, while (c) and (d) show results for $V = \Sigma_1^{-1}$ and $V = \Sigma_2^{-1}$, respectively. Each curve represents one random initialization.

Next, we consider a $V \in \mathbb{S}_+^N$ computed as follows. We run the time-series regressions (Eq. (5)) of $X$ on $\{\texttt{Mkt-RF, SMB, HML}\}$ and $\{\texttt{Mkt-RF, RMW, CMA}\}$, obtain the estimated residuals and compute their sample covariance matrices denoted by $\Sigma_1$ and $\Sigma_2 \in \mathbb{S}_+^N$. Then, we normalize $\Sigma_1$ and $\Sigma_2$ by dividing them by $tr(\Sigma_1)/N$ and $tr(\Sigma_2)/N$, respectively, in order for the normalized covariance matrices to have $tr(\Sigma_1) = tr(\Sigma_2) = N$, which equals the trace of $I_N$. We run Algorithm 1 for $V \in \{\Sigma_1^{-1}, \Sigma_2^{-1}\}$ and plot the objective function values $\phi_n$ for $V = \Sigma_1^{-1}$ and $V = \Sigma_2^{-1}$ in the subfigures (c) and (d) of Figure 4, respectively. We see that only 5 iteration steps are enough for the objective function values to converge for every random initialization. In the interest of space, we have omitted results on the numerical properties for $\eta \in \{0.1, 1, 20, 30, 40, 50, \cdots, 100\}$ since they are almost identical to Figure 4. In Appendix D, we present convergence results of the algorithm when applied to larger real-world data sets with $N \in \{25, 370\}$ and $T \in \{60, 240, 600\}$. Our findings show that the algorithm performs effectively across a range of realistic scenarios.

**Economic properties** We conduct Monte Carlo experiments using synthesized asset returns that follow the four factor model outlined in Lettau & Pelger (2020a). The factors $f_t$, loadings $\Lambda_i$ and residuals $\epsilon_t^i$ are generated from the normal distributions $\mathcal{N}(\mu_F, \Sigma_F)$, $\mathcal{N}(0, I_K)$ and $\mathcal{N}(0, \Sigma_E)$, respectively. We set $\Sigma_F$ as a diagonal matrix with values $(5, 0.3, 0.1, 0.03)$ and determine $\mu_F$ such that the Sharpe ratios $SR_F = (0.12, 0.1, 0.3, 0.5)$ where the $k$-th entry of $SR_F$ is defined as $SR_F^k = E(f^k)/\sqrt{var(f^k)}$. We consider two different experiment scenarios by setting $\Sigma_E$ to be either $\Sigma_1$ or $\Sigma_2$ defined earlier in this section. For each Monte Carlo experiment, we sample $f_t, \Lambda_i$ and $\epsilon_t^i$ for all $i \in \{1, 2, \cdots, 25\}$ and $t \in \{1, 2, \cdots, 60\}$ to generate asset returns by $X_{ti} = \Lambda_i^\top f_t + \epsilon_t^i$. For each scenario, we run 2000 Monte Carlo experiments and present their average of three performance measures: (i) $RMS_\alpha = \sqrt{\hat{\alpha}^\top \hat{\alpha}/N}$ measures the extent to which the estimated pricing errors are jointly zero. (ii) $RMS_\alpha^{\Sigma_\alpha} = \sqrt{\hat{\alpha}^\top \Sigma_\alpha^{-1} \hat{\alpha}/N}$ quantifies the distance between $\hat{\alpha}$ and $0$, in line with the motivation outlined in Section 2 through the use of $\Sigma_\alpha$ explained below. (iii) The maximum Sharpe

---

[1]Precisely, the suboptimality and the distance converge to tiny values smaller than $10^{-14}$.

ratio (SR) implied by the estimated factors measures the estimated factors' ability to approximate the the expected returns of the test assets. We compare economic performance of four estimators, namely, PCA, RP-PCA and two PCA-XC estimators written as PCA-XC$_p$ and PCA-XC$_{pr}$ for a range of $\eta \in [0.25, 100]$. PCA-XC$_p$ represents the PCA-XC estimator whose $V$ is set to the inverse of $\Sigma_\alpha = \Lambda \Sigma_F \Lambda^\top + \Sigma_E$ which is the population covariance matrix of the cross-sectional pricing errors $\alpha$ in the framework of cross-sectional regression (Cochrane, 2005). PCA-XC$_{pr}$ corresponds to a restricted case where $V$ is set to the inverse of $diag(\Sigma_\alpha)$. We apply the normalization to $V$ as explained above in this section.

Figure 5 exhibits the results obtained from the 2000 experiments where we can observe that PCA-XC$_p$ has the larger RMS$_\alpha$ than RP-PCA. This is because RP-PCA directly forces RMS$_\alpha$ to be small in its objective function. When it comes to RMS$_\alpha^{\Sigma_\alpha}$, PCA-XC$_p$ works the best with modest value of $\eta \in [3, 18]$, as it directly controls RMS$_\alpha^{\Sigma_\alpha}$ in its objective function, aligning with the motivation of the PCA-XC estimator. When $\eta$ is too small or too large, the difference between PCA-XC and RP-PCA becomes negligible since the second or first term, respectively, of the objective function $\phi$ is neglected, making either of the two important information for estimation is abandoned, effectively. In the third column, we can observe that PCA-XC$_p$ exhibits the highest Share ratio with statistical significance when the value of $\eta$ is moderate, which sheds light on how we could use PCA-XC and choose $V$ in order to obtain better implied Sharpe ratio. We can see that PCA-XC$_{pr}$ ranks second for all of the three performance metrics and that the results are similar for the two scenarios characterized by $\Sigma_E$ being equal to either $\Sigma_1$ or $\Sigma_2$.

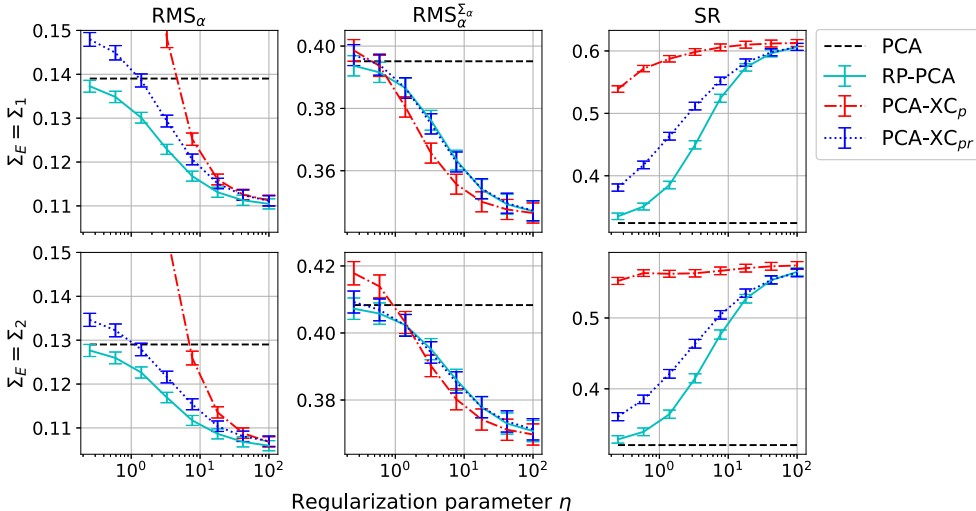

Figure 5: Economic properties of factor pricing model estimators. The results are average of results from 2000 Monte Carlo experiments and the vertical bars indicate 95% confidence intervals. Superior performance is indicated by lower values of RMS$_\alpha$ and RMS$_\alpha^V$ and higher values of SR.

Computation was executed on a laptop computer equipped with a Dual-Core Intel Core i7 3.3 GHz processor and 16 GB LPDDR3 RAM, operating on Ubuntu 18.04.6 LTS. Python 3.10.8 was utilized for all computations. The execution time for the experiment corresponding to Figure 4 was approximately 6 seconds, while the experiment for Figure 5 completed in under 8 hours.

## 6 CONCLUSION

We study one of the central problems in the field of finance and cast it into a problem of unsupervised learning with regularization. Our novel factor pricing model estimator is built on empirical observations and motivated by the need for model evaluation. This approach offers a broader and more comprehensive assessment of pricing errors used for model estimation, particularly accommodating scenarios with substantial cross-sectional correlations. We present an efficient algorithm for optimizing our proposed estimator.

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
