# OpenReview forum: "Principal Component Analysis for Cross-Sectionally Correlated Pricing Errors"
_ICLR.cc/2024/Conference — ICLR 2024 Conference Withdrawn Submission_

### Official Review · Reviewer_D5h2 · 2023-10-29

**Soundness:** 1 poor
**Presentation:** 1 poor
**Contribution:** 1 poor
**Rating:** 3
**Confidence:** 5

**Summary:**

The authors consider the problem of factor pricing for financial data. Factor models are very popular in financial literature for modelling dependencies between price returns. The authors propose a method to estimate the PCA in the case where the error is measured with a generalised norm $\| \cdot \|_{V}$.

The estimation method is based on alternating gradient descent, where the matrix decomposition factors are optimized directly.

Unfortunately, the paper does not offer anything new to the machine learning community, and I think there is not much for the financial literature as well. It is very well known that PCA can be solved by optimizing the factors in matrix decomposition with SGD. The questions that are actually interesting - choice of K, number of factors, and choice of matrix V that measures the distance, are conveniently silenced.

**Strengths:**

see summary

**Weaknesses:**

see summary

**Questions:**

see summary

---

### Official Review · Reviewer_mR2q · 2023-10-31

**Soundness:** 2 fair
**Presentation:** 2 fair
**Contribution:** 2 fair
**Rating:** 5
**Confidence:** 3

**Summary:**

The authors introduce an estimator called Principal Component Analysis for Cross-Sectionally Correlated Pricing Errors (PCA-XC) in the context of factor pricing models. This estimator is designed to identify a factor pricing model that effectively captures the time-series variation of asset returns while addressing the cross-sectional correlations of the errors observed in real-world data. Unlike existing estimators with analytically solvable minimization problems, the proposed PCA-XC relies on a challenging regularized minimization problem. To tackle this difficulty, the authors put forward an approximate algorithm based on the alternating least squares method for solving the new estimator.

**Strengths:**

Factor models for time series are not new and have been well-studied in the literature. The author claims that the proposed method can effectively capture the time-series variation and cross-sectional correlations among noises. However, their method does not compare with existing methods for the factor model of time series. Given that the proposed estimator is computationally difficult, it appears to lack significant evidence to support such a complicated method for the target problem.

**Weaknesses:**

The author reviews some methods for solving the factor model of time series using PCA. However, PCA is not the sole method available for addressing the target problem. Many other methods are designed to tackle this issue, such as [1] and [2]. These alternative methods are also capable of capturing time-series variations and addressing cross-sectional correlations in errors, and therefore, they should be included and compared within the article. Furthermore, the identifiability of the factor and factor loading is usually an important issue in factor models [1,2]. This issue should be addressed and discussed for the proposed method.

[1] Lam, Clifford, Qiwei Yao, and Neil Bathia. "Estimation of latent factors for high-dimensional time series." Biometrika 98.4 (2011): 901-918.

[2] Lam, Clifford, and Qiwei Yao. "Factor modeling for high-dimensional time series: inference for the number of factors." The Annals of Statistics (2012): 694-726.

**Questions:**

1.	Review more competing existing methods for solving the factor model of time series in Section 3.
2.	The details of factor models are not sufficiently demonstrated, for example, the identifiability of the factor and factor loading. Furthermore, the assumption of constraint (2) can be confusing, as it appears to contradict the objective of finding a model with smaller pricing errors.
3.	In Section 5, it would be convincing to include other competing methods, rather than only focusing on PCA-based methods, for estimating factors and factor loadings from the data matrix.

---

### Official Review · Reviewer_dcRp · 2023-11-01

**Soundness:** 3 good
**Presentation:** 3 good
**Contribution:** 2 fair
**Rating:** 3
**Confidence:** 3

**Summary:**

This paper considers PCA estimation of $F$ and $\Lambda$ by minimizing the sum of the following two terms:

(1) $ \frac{1}{NT} || \tilde{X} - \tilde{F} \Lambda^T ||_{F}^2$, which is the standard objective function for PCA, and

(2) $\frac{\eta}{N} || \bar{X} - \Lambda \bar{F} ^T ||_{V}$, which is an extra term that penalizes the pricing error.

This paper builds on Lettau & Pelgar (2020a,b) that considered the same problem with V being restricted to the identity matrix.

It is argued in the current paper that a general $V$ would be important to deal with correlated pricing errors. The presence of a general matrix $V$ creates a computational hurdle. Specifically, the estimator of Lettau & Pelgar (2020a,b) can be solved easily; however, the new estimator with a general $V$ does not have a simple closed-form characterization and is obtained via the alternating least squares. Proposition 4.1 provides conditions under which the proposed solutions are unique. Experiments are conducted to demonstrate the proposed estimator.

**Strengths:**

1. The paper builds on recent important work in financial econometrics and is written pretty well.

2. The presence of correlated pricing errors is quite plausible in asset pricing.

**Weaknesses:**

1. It is unclear what extent this paper makes a substantial contribution relative to Lettau & Pelgar (2020a,b). For example, the present paper does not provide any results regarding the statistical properties of the proposed estimator. On the contrary, Lettau & Pelgar (2020a) developed asymptotic results under both strong and weak factor regimes.

2. It is unclear how to choose $V$ in practice, not to mention the penalization parameter $\eta$. Without a guidance on these two crucial inputs, it would be hard for applied researchers to adopt the proposed method.

**Questions:**

1. What is the relationship between the global solutions to the original problem ($\mathcal{P}$) on page 6 and the solutions via the alternating least squares in Algorithm 1? Could they be different or they must be the same?

2. It is unclear what the uniqueness means in the statement of Proposition 4.1. It would be helpful to provide some clarifications regarding the implications of Proposition 4.1.

---

### Official Review · Reviewer_ihuY · 2023-11-06

**Soundness:** 3 good
**Presentation:** 2 fair
**Contribution:** 2 fair
**Rating:** 3
**Confidence:** 3

**Summary:**

This paper proposed a new factor pricing model that extends RP-PCA. A new optimization formulation with a seminorm constraint is introduced. The motivation of the method is obtaining a small test statistic when testing zero alphas.

**Strengths:**

The motivation is clear, and the methodology has limited novelty. They also proposed an algorithm with theoretical guarantee to solve the optimization problem.

**Weaknesses:**

1. While the motivation of the paper is small $|p|$ in a statistics test, why arbitrary $V$ is necessary? Any summary for all possible choices of $V$?

2. In what aspects the proposed method is better than RP-PCA? It is not clear.

3. The method is motivated by the asset pricing problem. Hence, it is necessary to prove its benefits for asset pricing in the experiments, as RP-PCA did.

4. I'm concerned that the topic of this paper does not have a broad audience in the machine learning community.

**Questions:**

I have listed my questions above.